# JLGBMLoc—A Novel High-Precision Indoor Localization Method Based on LightGBM

**DOI:** 10.3390/s21082722

**Published:** 2021-04-13

**Authors:** Lu Yin, Pengcheng Ma, Zhongliang Deng

**Affiliations:** School of Electronic Engineering, Beijing University of Posts and Telecommunications, Beijing 100876, China; mpc@bupt.edu.cn (P.M.); dengzhl@bupt.edu.cn (Z.D.)

**Keywords:** indoor localization, Wi-Fi fingerprint, denoising auto-encoder, JLGBMLoc

## Abstract

Wi-Fi based localization has become one of the most practical methods for mobile users in location-based services. However, due to the interference of multipath and high-dimensional sparseness of fingerprint data, with the localization system based on received signal strength (RSS), is hard to obtain high accuracy. In this paper, we propose a novel indoor positioning method, named JLGBMLoc (Joint denoising auto-encoder with LightGBM Localization). Firstly, because the noise and outliers may influence the dimensionality reduction on high-dimensional sparseness fingerprint data, we propose a novel feature extraction algorithm—named joint denoising auto-encoder (JDAE)—which reconstructs the sparseness fingerprint data for a better feature representation and restores the fingerprint data. Then, the LightGBM is introduced to the Wi-Fi localization by scattering the processed fingerprint data to histogram, and dividing the decision tree under leaf-wise algorithm with depth limitation. At last, we evaluated the proposed JLGBMLoc on the UJIIndoorLoc dataset and the Tampere dataset, the experimental results show that the proposed model increases the positioning accuracy dramatically compared with other existing methods.

## 1. Introduction

In recent years, location based service (LBS) has developed rapidly. However, due to severe signal attenuation and multipath effects, general outdoor positioning facilities (such as GPS) cannot work effectively in buildings [1]. Therefore, several types of indoor positioning technologies have been proposed, such as wireless local area network (WLAN), visible light, cellular networks and their combination technologies [2,3]. The indoor positioning based on Wi-Fi signals has the advantages of convenient deployment, low hardware cost and high real-time performance. However, Wi-Fi based indoor positioning faces the problem of the volatility of Wi-Fi signals and the high-dimensional sparseness of fingerprint [4]. This study focused on improving indoor positioning using a Wi-Fi fingerprint.

Generally, a Wi-Fi system consists of some fixed access points (APs) [5]. Mobile devices (such as laptops and mobile phones) that connect to Wi-Fi can communicate directly or indirectly with each other. Received signal strength (RSS) of the AP is usually used to pre-build a fingerprint database to infer the location of the mobile user. There are two stages in fingerprints positioning—the offline stage and the online stage [6]. The offline stage is to measure RSS readings of known locations from the surrounding access points and correlate them with these physical locations to build a fingerprint database. The collected data is the training set; In the online phase, the real-time sampled RSS vector of the target is compared with stored fingerprints for positioning, where the location of the best matched fingerprint is selected as the target location, the positioning result will be sent back to the requester.

Literature research faces two key problems in fingerprint based localization. Firstly, the observed RSS vectors contain a large number of missing values due to the obstruction of out-of-range APs, random noise, signal fluctuation or scanning duration [7], especially inside large buildings, such as shopping malls and hospitals, which results in extreme data sparsity. Traditional data dimensionality reduction methods, including principal component analysis (PCA) [8] and linear discriminant analysis (LDA) [9], treat all samples as a whole to find an optimal linear mapping projection with the smallest mean square error. But it has a poor reduction effect on complex data. With the development of neural networks, feature extraction and fusion have become more popular.

Another challenge is how to achieve high-precision and high-efficiency localization under multipath and noise fluctuations. The indoor propagation of Wi-Fi signals is easily affected by the human body, some obstacles, and walls, which affects the accuracy of fingerprint positioning. Traditional machine learning methods, including k-nearest neighbor (KNN) [10] and support vector machine (SVM) [11], are not effective in dealing with non-linear problems. Compared with these algorithms, the artificial neural network (ANN) [12] estimates the non-linear position from the input through the selected activation function and adjustable weights, and has the ability to approximate high-dimensional and highly nonlinear models. Notice that ANN is fully connected, the depth of the neural network is directly related to the complexity of its calculation, which may directly affect the accuracy of positioning results. In [13], a hybrid deep learning model (HDLM) is proposed to to enhance the localization performance of the existing Wi-Fi RSSI signal based positioning systems and reduce the positioning error, which uses RSSI heat maps instead of raw RSSI signals from APs. Hoang proposed a recurrent neural network for an accurate received signal strength indicator (RSSI) indoor positioning [14], using the results of different types of RNN, including long short-term memory (LSTM) [15] and gated recurrent unit (GRU) [16]. However, these algorithms still face challenges such as spatial ambiguity and RSS instability. In [17], a convolutional neural network (CNN) based indoor localization system with WiFi fingerprints is proposed, which has a 95% accuracy of floor-level localization on the UJIIndoorLoc dataset.

In summary, RSS based indoor positioning still faces the problem that noise and outliers affect high-dimensional sparse fingerprint data, and it is difficult to achieve high accuracy and high efficiency under multipath and noise fluctuations. To solve the above problem, this paper focuses on a novel feature extraction algorithm to reconstruct sparse fingerprint data to obtain better feature representation. Moreover, in order to reduce the space complexity and low training speed due to the pre-sorting algorithm of the existing gradient boosting model, a novel positioning model is introduced to disperse the processed fingerprint data into histograms and to divide the decision tree under the leaf-wise algorithm with depth limitation, which solves the problem of large space occupation and improves the calculation speed. The main contributions of this work are summarized as follows:(1)Aiming at the problem of extracting key features from sparse RSS data and reducing the influence noise and outliers of dataset, we propose a novel feature extraction algorithm, named joint denoising auto-encoder (JDAE), which reconstructs the sparseness fingerprint data for a better feature representation and restores the fingerprint data.(2)To achieve higher positioning accuracy under high efficiency, the LightGBM is introduced to the Wi-Fi localization by scattering the processed fingerprint data to histogram, and dividing the decision tree under the leaf-wise algorithm with depth limitation.(3)The proposed model is evaluated by the UJIIndoorLoc [18] and Tampere [19] datasets. The experimental results show that the proposed model is superior to traditional machine learning methods, the room-level positioning accuracy can reach 96.73% on UJIIndoorLoc, which is nearly 10% higher than the DNN method [20], and the floor-level positioning accuracy can reach 98.43% on Tampere, which is more predominant than current advanced methods.

The rest of this article is organized as follows—Section 2 introduces the background. Section 3 describes the architecture and the process of positioning based on our proposed model. In Section 4, we describe the preprocessing datasets, optimize the parameters of the model through experimental research and compare it with several benchmarks of positioning accuracy. Finally, we summarize the contribution of this work in Section 5.

## 2. Preliminary

### 2.1. Denoising Auto-Encoder

Auto-encoder is an unsupervised algorithm that automatically learns features from unlabeled data, which give a better feature description than the original data [21], and complete automatic selection of features, as shown in Figure 1. In [22], an AutLoc system is proposed to utilize an auto-encoder to improve the accuracy of indoor localization by preprocessing the noisy RSS by training the deep auto-encoder to denoise the measured data and then build the RSS fingerprints according to the trained weights. However, the positioning accuracy of this method can be further improved. Considering that datasets based on large buildings have strong sparsity, the output location information only depends on a small part of the dimensions of the input vector, which means the auto-encoder can effectively reduce the dimension of data, and the necessary feature information is retained. This conclusion will be confirmed in subsequent experiments.

Unlike the auto-encoder, part of the input data is “corrupted” during the training process of the denoising auto-encoder (DAE) [23]. In addition to having the properties of minimal refactoring errors or sparseness, auto-encoders can also be required to have the property of robustness to partial data corruption. The denoising auto-encoder is a kind of auto-encoder which increases the robustness of coding by introducing noise. For the input vector *x*, we first randomly set the values of some dimensions of *x* to 0 according to a ratio and get a damaged vector. The corrupted vector input is then given to the auto-encoder and the original lossless input *x* is refactored. The “corrupted” process is equivalent to adding noise. The core idea of DAE is to encode and decode the “corrupted” original data, and then recover it to reduce the noise of the data and improve the robustness of the model.

The principle is shown in Figure 2, where $f_{\theta }$ is the encoder, $g_{\theta^{\prime }}$ is the decoder, and $L(x,x^{\prime })$ is the loss function of the DAE network. The input data x is “corrupted” by noise according to the $q_{D}$ distribution. The current problem is to adjust the network parameters, by calculating $L(x,x^{\prime })$, to make sure the final output $x^{\prime }$ close to the original input *x*. In (1), *W* is the link weight from the input layer to the intermediate layer, and *b* is the bias of the intermediate layer. The signal $x_{0}$ is decoded by the decoding layer and transport to the output layer becomes *z*. In (2), $W^{\prime }$ is the link weight from the intermediate layer to the output layer, and $b^{\prime }$ is the bias of the output layer. $x^{\prime }$ is regarded as the prediction of *x*. We firstly randomly set the values of some dimensions to 0 according to a ratio to obtain a damaged vector $x_{0}$. Then the damaged vector is reconstructed to a lossless output $x^{\prime }$ as the input of the training model.
(1)$$z=f_{\theta }\left(Wx_{0}+b\right)$$
(2)$$x^{\prime }=g_{\theta^{\prime }}\left(W^{\prime }z+b^{\prime }\right)$$
(3)$$L\left(x,x^{\prime }\right)=||x-x^{\prime }{\left|\right|}^{2}.$$

### 2.2. Classification and Regression Tree

GBDT (Gradient Boosting Decision Tree) is an integrated learning of additive models based on regression trees [24]. The main idea is to continuously add weak learning functions and to perform feature splitting to grow a tree. In [25], Wang propose an algorithm named Subspace gradient boost decision tree (Subspace-GBDT) to obtain a strong classifier, which reduce the uncertainty caused by a single fingerprint, and the multiple fingerprints are based on the signal Subspace and RSSI, where the signal Subspace represents the characteristic representation of the received array signal. GBDT uses classification and regression tree (CART) as a weak learning function, which refers to a decision tree that uses a binary tree as a logical structure to complete linear regression tasks. The CART classification tree algorithm uses Gini coefficient instead of information gain. The smaller Gini coefficient, the better model features. Assuming that the dataset has K categories, the probability of the *k*-th category is $p_{k}$, the Gini coefficient expression of the probability distribution is:(4)$$Gini\left(p\right)=\sum_{k=1}^{K}p_{k}\left(1-p_{k}\right)=1-\sum_{k=1}^{K}p_{k}^{2}.$$

For the sample set *D*, assuming that the sample has *K* categories, the number of *k*-th categories is $C_{k}$, then the Gini coefficient expression of sample *D* is:(5)$$Gini\left(D\right)=1-\sum_{k=1}^{K}{\left(\frac{\left|Ck\right|}{\left|D\right|}\right)}^{2}.$$

According to value *a* of a certain feature *A*, divide *D* into $D_{1}$ and $D_{2}$, then under the condition of feature *A*, the Gini coefficient expression of sample *D* is:(6)$$Gini\left(D,A\right)=\frac{|D_{1}|}{\left|D\right|}Gini\left(D_{1}\right)+\frac{|D_{2}|}{\left|D\right|}Gini\left(D_{2}\right).$$

Therefore, for the CART classification tree, after the calculated Gini coefficient of each feature to the data set *D*, the feature *A* with the smallest Gini coefficient and the corresponding eigenvalue *a* are selected.

According to this optimal feature and optimal eigenvalue, the data set is divided into two parts $D_{1}$ and $D_{2}$, the left and right nodes of the current node are established at the same time. Until the Gini coefficient is less than the threshold, the decision tree subtree is returned, and the current node stops recursion. Notice that each time a tree is added, actually is to learn a new basic learner h(.) to fit the final prediction.

## 3. System Design

### 3.1. System Model

The LightGBM used in this paper is an improvement based on the algorithm of GBDT. Assume that the region of interest has *N* APs and *M* reference points (RPs), the RSS input set can be defined as $f=\{f_{1},f_{2},\ldots ,f_{M}\}$, and the corresponding location set is $l=\{l_{1},l_{2},\ldots ,l_{M}\}$. The GBDT algorithm can be regarded as an additive model composed of *K* trees:(7)$$g\left(f_{i}\right)=\sum_{k=1}^{K}h_{k}\left(f_{i}\right),i\in 1,2,\ldots ,M,$$
where $g(f_{i})$ represents the predicted output, which is the predicted position in the model, $f_{i}=\left\{F_{i1},F_{i2},\ldots ,F_{iN}\right\}$ is the RSS value set of the *i*-th sample, and $F_{ij}$ is the *j*-th eigenvalues (i.e., RSS value) of $RP_{i}$. Obviously, our goal is to make the predicted value $g(f_{i})$ of the tree group as close as possible to the true value $l_{i}=\left(x_{i},y_{i}\right)$, and have the largest possible generalization ability. According to the characteristics of the sample, each tree will fall into the corresponding leaf node and will correspond to a score. After completing training and getting *K* trees, the score corresponding to each tree is added to get the predicted value of the sample. In each iteration, on the basis of the existing tree, a tree is added to fit the residual between the prediction result of the previous tree and the true value. The integrated learner obtained by (t−1)-th iteration is $g_{t-1}\left(f\right)$, the focus of the *t*-th training is to minimize the loss function (8) with the square loss functions (9) and (10):(8)$$L\left(l,g_{t}\left(f\right)\right)=L\left(l,g_{t-1}\left(f\right)+h_{t}\left(f\right)\right)$$
(9)$$L\left(l,g_{t}-1\left(f\right)+h_{t}\left(f\right)\right)={\left(l-g_{t-1}\left(f\right)-h_{t}\left(f\right)\right)}^{2}={\left(r-h_{t}\left(f\right)\right)}^{2}$$
(10)$$r=l-g_{t-1}\left(f\right),$$
where *r* represents the residual. Each step of the GBDT algorithm needs to fit the residual of the previous model when generating the decision tree, and uses the fastest descent approximation method, which means the negative gradient of the loss function is used as the approximate value of the residual in the lifting tree algorithm. The negative gradient of the loss function of the *i*-th sample in the *t*-th iteration is:(11)$$r_{it}=-{\left[\frac{\partial L(l_{i},g\left(f_{i}\right))}{\partial g(f_{i})}\right]}_{g\left(f\right)=g_{t-1}\left(f\right)},$$
the residual obtained in the previous step is used as the new true value of the sample, and the data ($f_{i}$, $r_{it}$), i=1,2,..,N is used as the next tree training data to obtain a new regression tree, the corresponding leaf node area is $R_{jt}$, j=1,2,…,J, where *J* is the number of leaf nodes of the *t*-th regression tree. For leaf area *j*, we calculate the best fit value as:(12)$$\gamma_{jt}=\underset{\gamma }{\arg\min}\sum_{f_{i}\in R_{jt}}L(l_{i},g_{t-1}\left(f_{i}\right)+\gamma ).$$

Then, we update the strong learner and get the final learning function $g_{K}\left(f\right)$.The gradient boosting algorithm improve the robustness of data outliers through the loss function, which is greatly improved compared to the traditional machine learning algorithm.
(13)$$g_{t}\left(f\right)=g_{t-1}\left(f\right)+\sum_{j=1}^{J}\gamma_{jt}I\left(f\in R_{jt}\right).$$

GBDT can handle various types of data flexibly. However, due to the dependence between weak learners, it is difficult to train data in parallel, which results in relatively low operating efficiency of the model. Therefore, high dimensional data will increase the complexity of the model. LightGBM is a high-performance gradient boosting framework based on decision tree algorithm released by Microsoft in 2017 [26], which can be used in sorting, classification, regression and other machine learning tasks. LightGBM has been optimized on the GBDT algorithm to speed up the training of GBDT model without compromising accuracy.

In the improved gradient boosting model based on Wi-Fi positioning, XGBoost [27] uses the pre-sorting algorithm to reduce the amount of calculation to find the best split point. But it still needs to traverse the positioning data set during the node splitting process, which increases the space complexity and training speed. Compared with XGBoost, LightGBM uses the histogram algorithm to process the positioning data set and the leaf-wise split strategy in the process of Wi-Fi-based positioning, which solves the problem of large space consumption due to pre-sorting and improves the calculation speed.

Firstly, in our positioning fingerprint, AP${}_{j}$ is regarded as the j-th feature of fingerprint data, and $F_{j}=\left\{F_{1j},F_{2j},\ldots ,F_{Mj}\right\}$ is defined as a set of eigenvalues contained in AP${}_{j}$. Then, a histogram decision tree algorithm is imported to discretize $F_{j}$ into a histogram with a width of *k*. Instead of the traditional pre-sorting idea, each of these precise and continuous values is divided into a series of discrete domains. The histogram accumulates the required statistics according to the discrete value, and traverses to find the best positioning AP feature and the corresponding eigenvalue as the segmentation point. No additional storage of pre-classification results is needed, only discrete values of features can be saved, and memory consumption can be reduced to one-eighth of the original value. The histogram is shown in Figure 3.

Considering the high-dimensional sparsity of the fingerprint data, the features represented by many APs are mutually exclusive, which means they usually don’t take non-zero values at the same time. According to the exclusive feature bundle (EFB) algorithm of LightGBM, the complexity of fingerprint feature histogram construction changes from O(data∗feature) to O(data∗bundle), and bundle<<feature, which greatly accelerates the training speed of the gradient boost model without affecting the positioning accuracy.

Secondly, the traditional decision tree splitting strategy is to use level-wise to find the best positioning AP feature and the corresponding feature value as the split point. However, the AP feature of the same layer are treated indiscriminately, and many APs have lower split gain, which brings unnecessary cost. Therefore, a leaf-wise algorithm with depth limitation is used to find the feature with the largest split gain, which means the best feature of the fingerprint data can be found from all current leaves, and then split, to reduce more errors and obtain better accuracy. As shown in Figure 4, compared with level-wise, leaf-wise reduce more errors and get better accuracy when the number of splits is the same; However, when leaf-wise grow deeper, which causes decision tree over-fitting. Therefore, the maximum depth limit is added to prevent over-installation and ensure high efficiency.

### 3.2. Feature Extraction Algorithm

Since each iteration of the gradient boosting algorithm adjusts the sample according to the prediction result of the previous iteration, as the iteration continues, the bias of the model will continue to decrease, which leads to a model more sensitive to noise. In the indoor positioning dataset, the outliers caused by multipath signals and NLOS will have an impact on the training of the database.

To solve the problem of extracting key features from sparse RSS data and reducing the influence noise and outliers of a dataset, we propose a feature extraction algorithm, called the joint denoising auto-encoder (JDAE), aiming to extract key features from sparse RSS data and reduce the influence of noise and data outliers. Considering the sparseness of our positioning dataset, the input dimension is mostly 0. If we use DAE directly for a certain probability of input zeroing, it is probably that the dimension of our zeroing itself is 0, which make it ineffective. Thus, we add an auto-encoder in front of DAE for feature extraction, reducing the sparseness of the dataset. The feature output obtained by the auto-encoder is almost non-zero value for each dimensional, which is better for probability zeroing through DAE.

The architecture of JDAE is shown in Figure 5. From input layer to feature layer is the part of auto-encoder, *x* means the input RSS data, $h^{\left(1\right)}$ is the hidden layer of auto-encoder, and *f* means the feature data processed by the auto-encoder. This part is to extract key features from sparse Wi-Fi data.

After getting reconstructed features, next part is to introduce denoising auto-encoder to reduce the influence of noise and data outliers. The denoising auto-encoder randomly partially use the damaged input to solve the identity function risk, to make the auto-encoder denoised. The dropout in the auto-encoder network refers to randomly letting the weights of some hidden layer nodes of the network not work during model training. We apply dropout layyer to the input layer instead of the hidden layer. The damage ratio generally does not exceed 0.5, and Gaussian noise can also be used to damage the data. The feature is robustly obtained from the damaged input, which can be used to restore its corresponding noise-free fingerprint data. The “⊗” part in the feature layer means “corrupted” features according to our setting, and $h^{\left(2\right)}$ is the hidden layer of denoising auto-encoder. After processing dataset by JDAE, the output layer *d* is imported to the LightGBM model.

### 3.3. System Architecture

The positioning method based on LightGBM is divided into two stages, the offline training stage and the online positioning stage. In the training stage, the RSS of each predefined RP is collected in the database, and RP${}_{i}$ has a corresponding fingerprint vector $f_{i}$={$F_{i1}$, $F_{i2}$, …, $F_{iN}$} at its location $l_{i}$($x_{i}$, $y_{i}$, $R_{i}$), where *N* is the number of available features (i.e.RSS) of all APs. Note that $x_{i}$ and $y_{i}$ represent position coordinates, which are different from the meaning in the feature extraction diagram above, $R_{i}$ is the corresponding room ID. Considering using the large building dataset, the algorithm complexity of the location regression prediction is too high, and it is difficult to find some benchmarks to compare. So we changed the positioning problem to a room classification problem, which has the advantage of reducing the complexity of the algorithm and comparing it with existing advanced methods. The coordinates ($x_{i}$, $y_{i}$) are not used here as output, only room IDs are used. After standardizing the dataset, the proposed JDAE is introduced, aiming to extract key features from sparse RSS data and reduce the influence of noise and data outliers. And then, LightGBM is imported to classify the processed data, and adjust the input parameters according to the results to obtain the optimal model; In the online stage, the proposed model will localize each location by matching the received fingerprinting measurement and sending back the room ID to the mobile user. The detailed algorithm is shown in Figure 6. The mapping relationship between the location and the Wi-Fi signal data is learned through LightGBM.

The idea of training the processed dataset is to transform the positioning problem into a multi-classification problem through position discretization, and each position corresponds to a category. Then, the samples are trained and the results of each decision tree are fused to get the final classification result. The steps of using the algorithm to train the fingerprint are as follows:(1)Firstly, a certain location is selected as the sampling point, the Wi-Fi fingerprint data are collected as all the characteristics of the sample. The histogram method algorithm is used to discretize the eigenvalues of the sample into *K* integers, and construct a histogram of width *K* for each feature. Then, according to the discrete value of the histogram, each AP point is used as the feature of the dataset, and the AP point corresponding to the minimum loss function value and the corresponding eigenvalue is calculated as the best split point for each iteration;(2)In order to prevent the built fingerprint database model from being too complicated and over-fitting, it is necessary to limit each split of the node. Only when the gain is greater than the threshold, the split is performed and when a tree reaches the maximum depth, it stops continuing to split;(3)When generating a decision tree, the gradient boost algorithm is used to make the predicted result continuously approach the real result, and offline training is completed through the learning of multiple decision trees. In online positioning stage, each testing Wi-Fi data is normalized, and sent to the trained multi-classification model for positioning.

## 4. Experiment Evaluation

### 4.1. Data Preprocessing

The UJIIndoorLoc [18] dataset used in this paper covers three buildings of Jaume I University, with four or five floors and an area of nearly 110,000 square meters. It can be used for classification (for example, actual building and floor identification) or regression (estimation of longitude and latitude). It was created by more than 20 different users and 25 android devices. The database consists of 19,937 sets of training data and 1111 sets of testing data. As shown in Table 1, the 529 attributes contain RSS values, the coordinates of the locations and other useful information. Each Wi-Fi fingerprint can be characterized by the detected APs and corresponding RSS values. One Wi-Fi fingerprint consists of 520 intensity RSS values.

However, we found that all the room ID of the given testing set is 0, which means that the testing set can only achieve floor positioning, not room-level positioning. Therefore, we consider dividing the training set directly into training sets and testing sets. Using the K-fold cross-validation [28] method, we firstly divide the dataset into *K* subsets of mutual exclusions of the same size by layered sampling, and then each time, the K−1 subset of them are used as a training set, and the remaining one as a testing set, so that we can get the K-group training/testing set, the *K*-time model can be learned, and the average of *K* test results as an evaluation result. Considering the selection of K values, if there is a small-scale dataset, usually choose 5, which means 80% of the data set as a training set, 20% as a testing set; for large-scale data of tens of thousands of magnitude, the *K* value usually takes 10, 20, or 50. Here we select 20, and each iteration, the number of testing sets is about 1000.

On UJIIndoorLoc dataset, the value of the input RSS data varies from −104 dbm to 0 dbm, and is nomalized for model training. In [29], different data representations of RSS fingerprints may affect the success rate and error. For any AP that is not detected in a measurement, its RSS value is marked as 100 dbm, and we denote these RSS values as 0.
(14)$$\mathrm{Normalized}=\left\{\begin{array}{c} 0,RSS_{ij}=0\\ {\left(\frac{RSS_{ij}-\min}{-\min}\right)}^{\beta },otherwise, \end{array}\right.$$
where *i* is the AP identifier, RSS${}_{ij}$ the *j*-th RSS value of $RP_{i}$, the minimum value is the lowest RSS value considering all fingerprints in the database and the AP, and β is a mathematical constant. The normalization changes the range of values for each feature to [0,1] by scaling. The β value can be set to 1. The results in [29] show that the normalized data tend to express the RSS value with the best performance, and tame the fluctuating RSS signal. Therefore, the normalized data are used to represent the Wi-Fi fingerprint in this paper.

The Tampere dataset covers two buildings of Tampere [19] University of Technology. In the first building, there are 1478 sets of training data and 489 sets of testing data. 312 attributes include Wi-Fi fingerprints (309 AP) and coordinates. The intensity value is expressed as a negative integer ranging from −100 dBm to 0 dBm. The Wi-Fi fingerprint consists of 309 intensity values. In the second building, there are 583 sets of training data and 175 sets of testing data, including 357 attributes of Wi-Fi fingerprints (354 AP) and coordinates. The Tampere dataset uses floor height as floor representation instead of floor number. In this experiment, adjusted optimal model is verified with the Tampere to test the performance of JLGBMLoc.

The experiment is equipped with a laptop with Intel i5-6300 CPU, using python-3.7.6 on tensorflow environment to implement the model building. The parameters used in the initial optimization are shown in Table 2.
(15)$$MSE=\frac{1}{n}\sum_{i=1}^{n}{\left(y_{i}-y_{i}^{\prime }\right)}^{2}.$$

The loss function is the mean square error (MSE). The training batch is set to 60, and the patience parameter in the early stop is set to 5, feature fraction takes 0.8, which is equivalent to the learning rate. After one iteration, the weight of the leaf node will be multiplied by the coefficient. The purpose is to weaken the influence of each tree to make sure the later decision tree has more learning space.

### 4.2. Performance Evaluation of JDAE

The performance of the model is evaluated by comparing the performance of the model with the state-of-the-art methods. Two datasets, UJIIndoorLoc and Tampere, are used for experimental research. The ratio of the number of correct matching positions to the total number of positions is used as the accuracy rate to evaluate the effects of each method. The accuracy in this work is defined as follows, $N_{A}$ means correctly predicted number of samples, and *N* means total number of samples.
(16)$$Accuracy=\frac{N_{A}}{N}.$$

Firstly, we use UJIIndoorLoc to optimize the parameters of the model, the adjusted optimal model is verified with the Tampere dataset to test the performance of JLGBMLoc with different datasets. We firstly train the model in floor positioning, and then use the trained model to test room positioning accuracy.

The fixed parameters of LightGBM and default values is given in Table 2, and used alone to train UJIIndoorLoc without feature extraction. As shown in Figure 7, after completing the iteration, the loss function value reaches 0.51. The positioning accuracy of testing data finally reached 91.04%, which is nearly 10% higher than the DNN method in floor-level positioning [20]. The running time of LightGBM is 5.5 s, the speed of which is almost two times higher than XGBoost [27].

Then, different auto-encoder models are built to evaluate the best performance. The comparison result is shown in Figure 8. When the hidden layer and output layer are set to 128 and 64 respectively, the floor success accuracy rate reaches the highest 95.59%. Therefore, for auto-encoder, we choose the hidden and output layer to be 128 and 64.

The weights of the denoising auto-encoder is initialized. If the weight of the network is initialized too small, the signal will gradually shrink during transmission between layers and it will be difficult to produce any effect. The initialization method will automatically adjust the most appropriate random distribution according to the number of input and output nodes of a certain layer of network, which is to make the weight meet the 0 mean value. Assuming the number of input nodes in input dimension and the number of output nodes in output dimension, the variance of uniform distribution is 2/(input+output), and the form of random distribution can be uniform distribution. Here we make the input and output dimensions equal. The CDF (cumulative distribution Function) of two methods are shown in Figure 9. Compared with single LightGBM model, the accuracy has been further improved. Not only that, our JDAE method is 6% more accurate than the method using a single auto-encoder in [30].

### 4.3. LightGBM Parameter Optimization

Firstly, we tune $num_{-}leaves$, which is an important parameter to improve accuracy. Maximum depth represents the depth of the set tree; the greater the depth, the greater possibility of overfitting. Due to the leaf-wise algorithm used by LightGBM, number of leaves is used when adjusting the complexity of the tree. The approximate conversion relationship is:(17)$$\mathrm{num}\_\mathrm{leaves}\leq 2^{\left(\max\_\mathrm{depth}\right)}-1.$$

Secondly, when there is not enough training data, or over-training, it results in overfitting. As the training process progresses, the complexity of the model increases and the error on the training data decreases, but the error on the testing set increases. The over-fitting of the model on the training dataset is reduced by constraining the $lambda_{-}l1$ and $lambda_{-}l2$ norm of the parameters, which effectively prevent the model from over-fitting. After adjusting our model, $lambda_{-}l1$ and $lambda_{-}l2$ is taken as 0.01, and the optimal accuracy rate in floor classification reaches 97.07%. And then, learning rate is adjusted. In order to make the gradient improvement model have better performance, the value of the learning rate needs to be set in the appropriate range. The learning rate determines how quickly the parameters move to the optimal value. If the learning rate is too large, it is likely to cross the optimal value, but if the learning rate is too small, the optimization efficiency may be too low and the algorithm cannot converge. The initial learning rate is 0.05. When the learning rate is 0.6, the accuracy reaches 99.32%, and when it gets to 1, the accuracy drops to 95%. Therefore, the learning rate is set to 0.6. The results analysis is shown in Figure 10.

When locating the rooms, we changed the number of classes (considering 15 floors and about 250 rooms). In Figure 10, the learning rate, the $num_{-}leaves$ and other parameters are consistent with the floor positioning, the highest accuracy is also achieved in the room positioning. Here, we consider each floor as a group, and the corresponding room on the floor can be counted as an element of this group.Therefore, the selection of floor positioning parameters can be largely consistent with that of room positioning.

Finally, we optimize the parameter min_split_gain, which means the minimum gain for splitting the decision tree. After testing, the optimal parameter is 0.02. The parameters of the model is shown in Table 3. We use the optimized model to position and randomly selected about 1000 sets of data covering 50 rooms in the UJIIndoorLoc dataset for testing. We compare JLGBMLoc with DNN [20], CNNLoc [17] and LightGBM. The room success rate comparison is shown in Figure 11, and the room accuracy rate reaches 96.73%. As the complexity of room-level positioning will be much higher than floor positioning, the accuracy will be slightly reduced. Experiments show that our proposed model can achieve room-level positioning, and the accuracy is more predominant than current advanced methods.

### 4.4. Model Comparison

The performance of the model is evaluated by comparing JLGBMLoc with several state-of-the-art methods. Not only that, we test the accuracy of the position in Tampere. The regression of deep learning cannot be judged by the accuracy of classification problems. We calculate the MSE to detect the deviation between the predicted and true values of the model, and the MSE of coordinate regression is 4.22. Considering that regression prediction does not have a good baseline comparison, we used the height classification of the Tampere dataset to achieve floor positioning and compared it with current advanced methods. The initial accuracy rate is 95.45%, which is the parameter used before testing UJIIndoorLoc. After changing the parameter num_class to 5, and $lambda_{-}l1$ to 0.02 on Tampere, the accuracy rate increased to 98.43%. Benchmark methods include KNN [10], 13-KNN, DNN [20], CNN and CNNLoc [17]. The model comparison is shown in Figure 12. The floor success rate of JLGBMLoc on UJIIndoorLoc is 99.32%, which on Tampere is 98.43%. The results show that the performance of our model is better than other benchmarks, which proves its high accuracy and scalability in different scenarios and datasets.

## 5. Conclusions

In this paper, we proposed a novel indoor positioning method, named JLGBMLoc. A novel feature extraction algorithm was proposed to reconstruct the sparseness fingerprint data, and LightGBM was introduced to the Wi-Fi localization. We evaluated the proposed JLGBMLoc on the UJIIndoorLoc dataset and the Tampere dataset; the experimental results showed that the proposed method has a room-level positioning accuracy of 96.73%, a floor-level positioning accuracy of 99.32% on the UJIIndoorLoc, and a floor-level accuracy of 98.43% on the Tampere. Experimental results proved that the proposed JLGBMLoc increases the positioning accuracy dramatically compared with other existing methods.

## Figures and Tables

**Figure 1 sensors-21-02722-f001:**
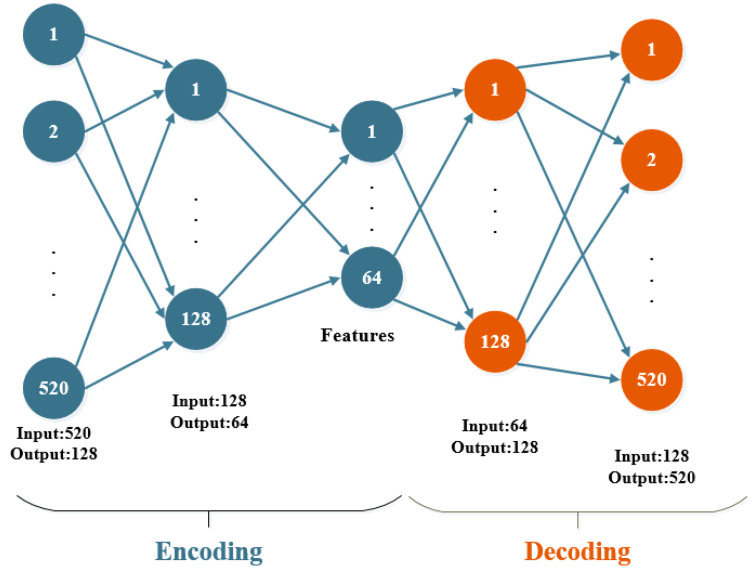
Auto-encoder Structure Chart.

**Figure 2 sensors-21-02722-f002:**
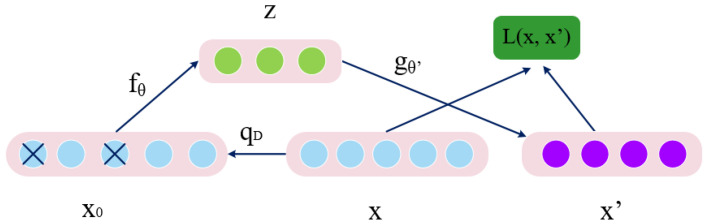
Denoising Auto-encoder Structure Chart.

**Figure 3 sensors-21-02722-f003:**
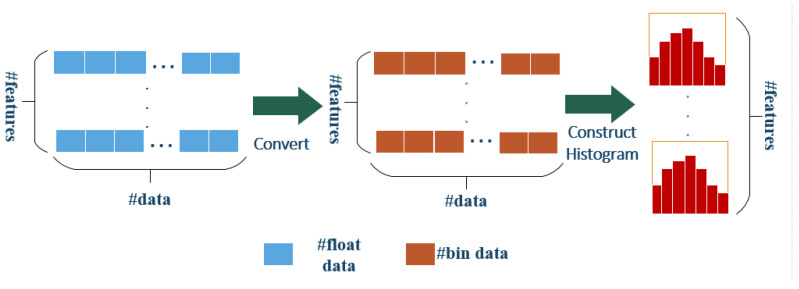
Histogram algorithm of LightGBM.

**Figure 4 sensors-21-02722-f004:**
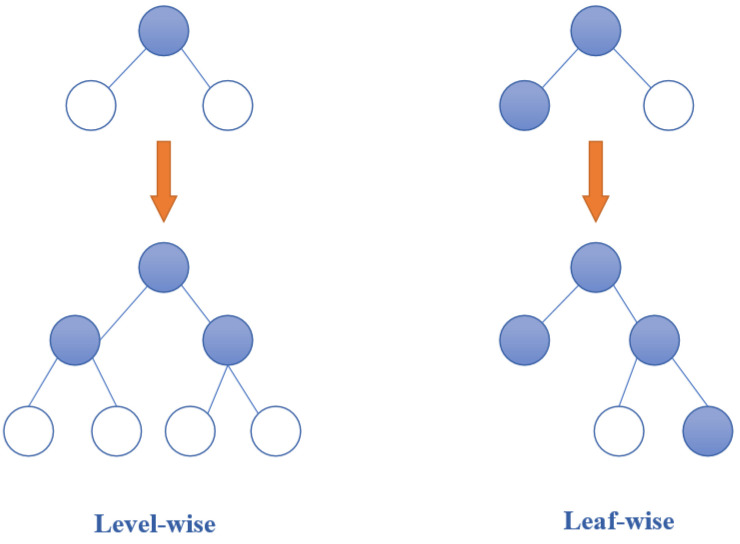
The generation strategy of tree.

**Figure 5 sensors-21-02722-f005:**
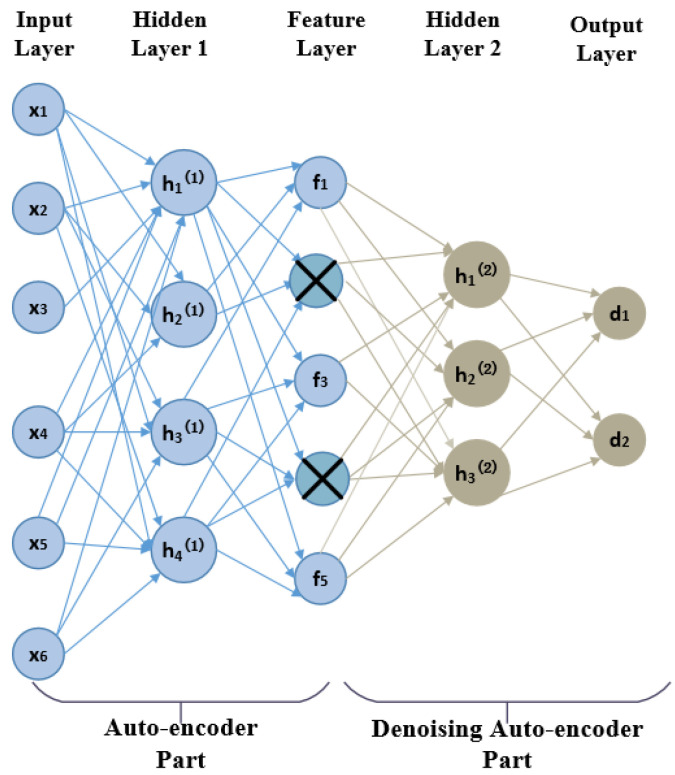
The architecture of the joint denoising auto-encoder (JDAE).

**Figure 6 sensors-21-02722-f006:**
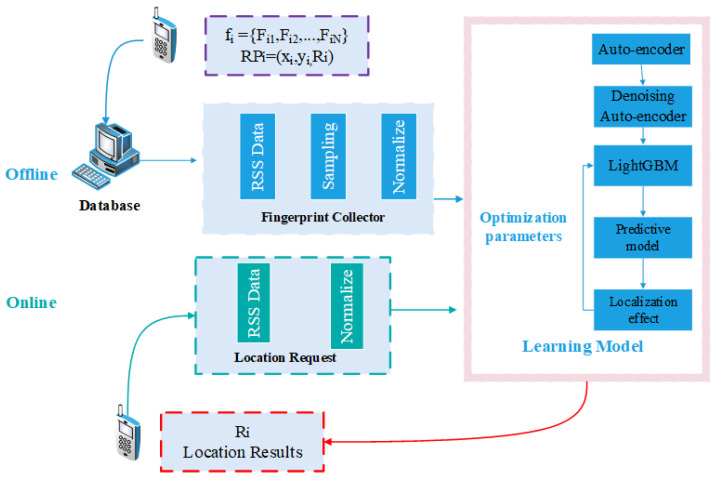
System Architecture.

**Figure 7 sensors-21-02722-f007:**
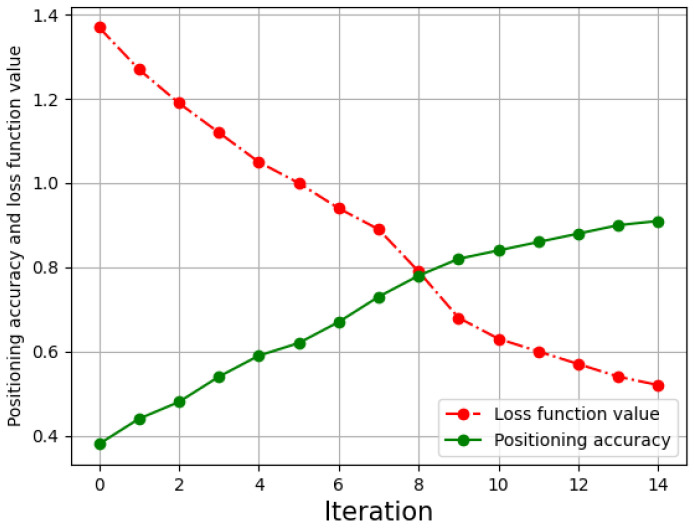
LightGBM localization without feature extraction.

**Figure 8 sensors-21-02722-f008:**
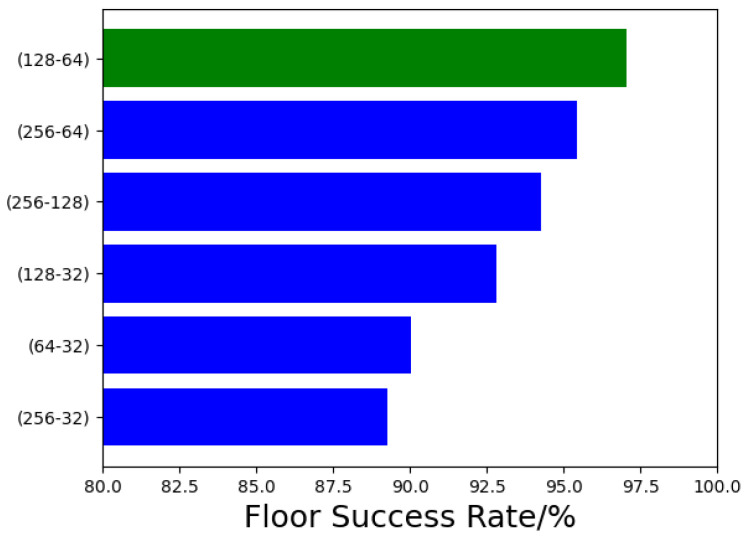
Effect of different auto-encoder models on floor localization.

**Figure 9 sensors-21-02722-f009:**
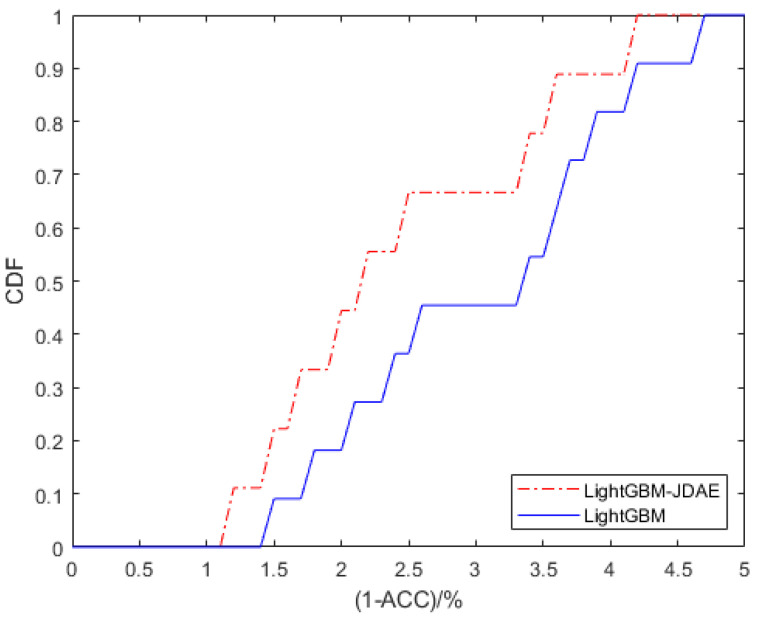
CDF comparison of the two methods.

**Figure 10 sensors-21-02722-f010:**
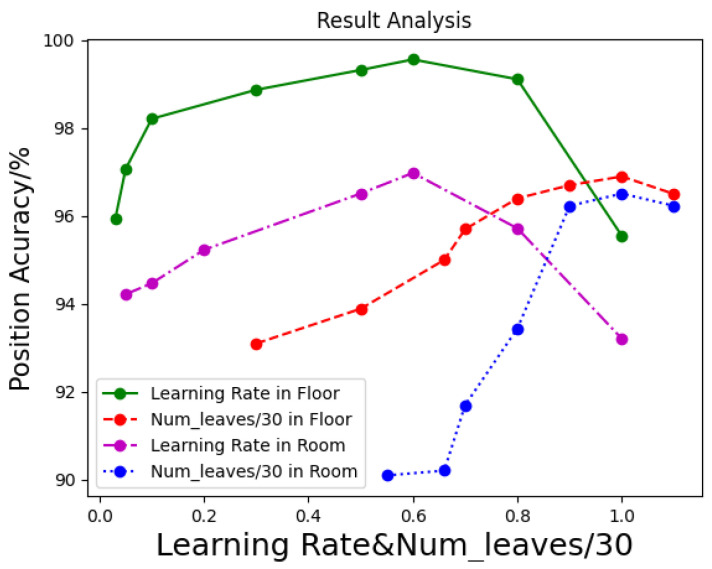
Optimization of learning rate and $num_{-}leaves$.

**Figure 11 sensors-21-02722-f011:**
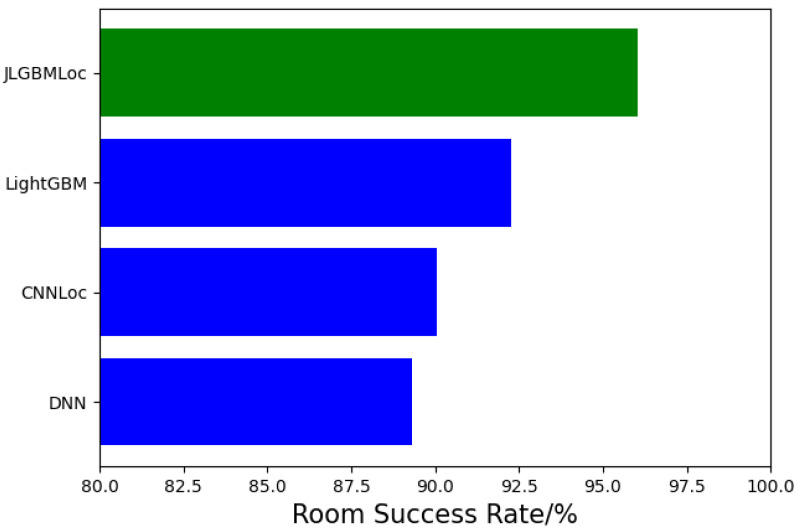
Room success rate comparison of different models.

**Figure 12 sensors-21-02722-f012:**
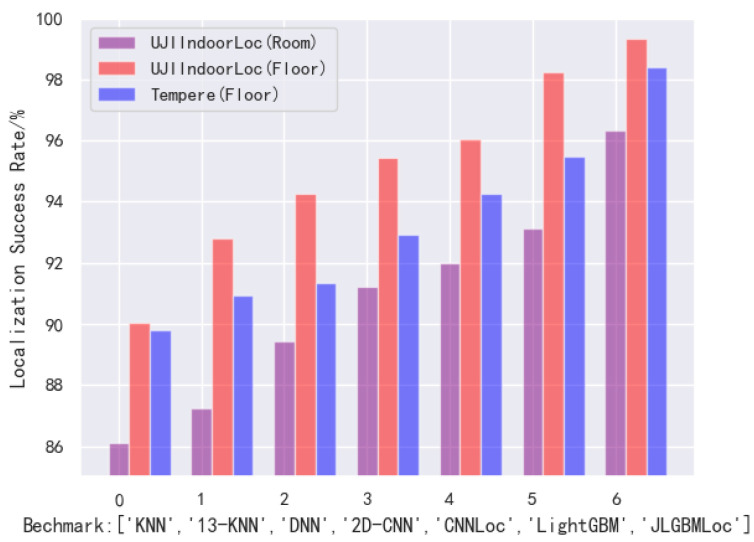
Localization result comparison on two datasets.

**Table 1 sensors-21-02722-t001:** The Important Information of UJIIndoorLoc.

Attribute	Information
001	(AP001): Intensity value for AP001
…	…
520	(AP520): Intensity value for AP520
521	Longitude
522	Latitude
523	Floor ID
524	Space ID

**Table 2 sensors-21-02722-t002:** Parameter Settings.

Parameter	Setting
Subsample	0.8
Lose	MSE
Early stoping patience	5
Batch size	60
Feature fraction	0.8

**Table 3 sensors-21-02722-t003:** Parameter Tuning.

Parameter	Value
learning rate	0.6
$lambda_{-}l1$	0.01
$lambda_{-}l2$	0.01
$num_{-}leaves$	30
$max_{-}depth$	5
num_classes	250
min_split_gain	0.02

## Data Availability

The UJIIndoorLoc dataset and the Tempere dataset can be found here:http://indoorlocplatform.uji.es/databases/all/ (accessed on 3 November 2016). The UJIIndoorLoc dataset covers three buildings of Jaume I University, with 4 floors and an area of nearly 110.000 square meters. It was created in 2013 by means of more than 20 different users and 25 Android devices.

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
