# Peer review of "JLGBMLoc—A Novel High-Precision Indoor Localization Method Based on LightGBM"

_sensors, 2021, doi:10.3390/s21082722_

Round 1

Reviewer 1 Report

  • Page 3 of 15 line 110 :

“The real raw data can reduce the noise of the data and improve the robustness of the model.”

How can raw data reduce the noise of the data to improve the robustness of the model?  It may need rephrasing.

  • Page 8 of 15 , line 220 Figure 6.System Architecture , in the online part, for the location request the response is coordinate (xi,yi) .Do the authors get this exact coordinate by classification or regression? Does the method follow the same steps to get room level and exact coordinate position? Need clarity.
  • Page 9 of 15 Table 1: The Information of UJIIndoorLoc. There are only 524 attributes but in line 249 it says 529 attributes. There is inconsistency?
  • Page 9 of 15 Line 255 : What is the value of the mathematical constant B used ? in the cited paper [26] the authors  use ‘e’ .
  • There is need for more results and analysis. For example the model analysis in page 13 of 15 , line 320 is only one and it is only for floor level. There need more result analysis for exact localization, to fit to the title of the manuscript.
  • Page 13 of 15 line 336 : floor level accuracy of49% on Tampere but it is 95.45% as in page 13 of 15 line 327 why is the difference?
  • As clearly stated in No. 246, the UJIndoorLoc datasets are suitable
    for only building and  oor level classi cation positioning tasks. This
    implies that some data manipulation need to be done in order to use
    both the 19937 training and 1111 validation datasets of UJIndoorLoc for
    space/room-level classi cation based experiment. With nothing of such
    presented in the document as concerning the room-level positioning, it is
    suspected that either a portion of the same training data that was used
    in training the model was used for the room-level testing experiment or
    the training dataset was partitioned in parts for training, validating and
    testing the proposed model.
  • ˆ Inconsistency in table references by the alternating use roman numerals
    and decimal numbers. In the case where the room-level positioning was
    performed on a portion of the same data that was used in training the
    model, then the reported positioning performance is not very reliable.
  • According the statement, "We use the optimized model to position, ran-
    domly select about 1000 sets of data covering 50 rooms in the UJIIndoor-
    Loc dataset for testing." in No. 312-313, there is a highly probability that
    the data for room-level testing was selected from the UJIndoorLoc training dataset. This is because no data manipulation of any sort was presented in the document as concerning the room-level positioning, and there are approximately only 13 rooms in the UJIndoorLoc 1111 validation/testing dataset which covers rooms absent in the 19937 training dataset. Given that the proposed technique solves the indoor positioning problem as a classi cation problem as stated in No.222-223, it cannot correctly predict such room positions and hence will imply that the present results is falsed or has not been clear explained.
  • ˆ Considering maxdepth = 5, the results of Equation 17 does not match
    that which is displayed in the Table 3
  • ˆ The statement " Not only that, we test the accuracy of position regres-
    sion in Tempere" at No. 324, con ict with the statement at No. 222-223.
    Probably, given that the  oor heights in Tampere being discrete and not
    continuous, it is still possible to treat the positioning problem as a classi-
     cation problem. This can probably eliminate the ambiguity in the way
    the proposed technique is to solve the indoor positioning problem.
  • Other Comments
    The system design and experiment evaluation seem not correlate well in-terms of the type of positioning the system is being designed to perform (subarea or point positioning). This is so because the system design seems to describe a 2D (longitude,latitude) point positioning system whereas the experiment evaluation focuses on subarea positioning of  oor and room level. Furthermore, in the case the speci ed 2D coordinates in the system design represented  floor and room positions, the experiment evaluation deviated by treating the  floor and room positioning independently. Given that  oor and space/room labels are not unique to a building or  floor respectively in the UJIndoorLoc dataset, the performance accuracies presented in this document for the independent  at flat-level-based positioning might not be all that accurate.

Author Response

Thank you very much for your caring review of our manuscript and for pointing out our shortcomings and errors in the experimental process, methods, and syntax. We have corrected the manuscript in response to all the above points,and uploaded a point-by-point response as a word file.

Reviewer 2 Report

The authors present a fingerprinting based (RSS) system that is capable of floor detection as well as room classification. The proposed system utilizes gradient boosting (LightGBM) combined with autoencoder based data preprocessing to overcome sparsity of the fingerprinting data. The system is evaluated on two public fingerprinting datasets.

Especially the benchmark results that are presented in the study are impressive. The proposed model outperforms existing solutions (mostly based on deep learning) by a considerable margin.

However, several aspects regarding the overall presentation should be improved:

  • Related work (background) is only weakly described. No distinction is made between device-based and device-free fingerprinting although studies are cited from the latter one [25].
  • In general, related work is only briefly mentioned in the introduction. Gradient boosting for Wifi-fingerprinting or studies on dimensionality reduction to overcome fingerprint sparsity are missing. Instead, a study on DNN for integrity monitoring [13] is cited, which seems out of scope, since several other studies using deep learning for fingerprinting based indoor localization are present, which could have been cited instead (e.g. in the running or past Special Issues of Sensors e.g. https://www.mdpi.com/journal/sensors/special_issues/Indoor_Positioning_Systems). Therefore, I suggest that the authors thoroughly revise the related work section.

In general, the presentation of the system is mostly clear, however, I have some suggestions that should be considered:

  • The concept of learning rate (of gradient boosting) should be introduced when giving the background on that topic. Especially, since the term is ambiguous to the learning rate of neural network learning (autoencoder).
  • It is not clear, how the model is “trained in floor positioning and then used to test room positioning accuracy” (275-276). The classes should differ to the reviewer’s understanding, which would require retraining of the model (successively building the gradient boosted weak learners). Is it possible, that only the parameters used in floor detection are kept for building the room classification model?
  • The evaluation of room classification is quite unclear. For several floors of the UJI dataset fingerprints were only collected in the halls (no rooms present). The authors should describe in more detail, how rooms have been chosen during the evaluation and report basic information such as the size of the rooms etc.
  • During parameter tuning, the authors mention lambda_1 and lambda_2, which are not previously introduced when describing the background on LightGBM.
  • The data preprocessing utilized in this study was originally investigated in “Comprehensive analysis of distance and similarity measures for Wi-Fi fingerprinting indoor positioning systems” by Torres-Sospedra et al. and not as cited by [26], which only applied it. Authors should correct this.

Additionally, we have a few minor remarks:

  • The resolution of Figure 6 is too low.
  • The name of second dataset is Tampere instead of Tempere.
  • The claimed floor success rate for the Tampere dataset should match (95.45% in line 327 and 95.49% in line 336).

Author Response

(The authors gave the same response as above.)

Round 2

Reviewer 1 Report

The authors have addressed reviewers' comments satisfactorily.

The presentation and English should be further improved.